# Combining Long Short Term Memory and Convolutional Neural Network for Cross-Sentence $n$-ary Relation Extraction

**Angrosh Mandya**                                    ANGROSH@LIVERPOOL.AC.UK
**Danushka Bollegala**                              DANUSHKA@LIVERPOOL.AC.UK
**Frans Coenen**                                        COENEN@LIVERPOOL.AC.UK
**Katie Atkinson**                                        KATIE@LIVERPOOL.AC.UK
*Department of Computer Science, University of Liverpool*
*Liverpook, UK*

## Abstract

We propose in this paper a combined model of Long Short Term Memory and Convolutional Neural Networks (LSTM_CNN) that exploits word embeddings and positional embeddings for cross-sentence $n$-ary relation extraction. The proposed model brings together the properties of both LSTMs and CNNs, to simultaneously exploit long-range sequential information and capture most informative features, essential for cross-sentence $n$-ary relation extraction. The LSTM_CNN model is evaluated on standard dataset on cross-sentence $n$-ary relation extraction, where it significantly outperforms baselines such as CNNs, LSTMs and also a combined CNN_LSTM model. The paper also shows that the LSTM_CNN model outperforms the current state-of-the-art methods on cross-sentence $n$-ary relation extraction.

## 1. Introduction

Research in the field of relation extraction has largely focused on identifying binary relations that exist between two entities in a single sentence, known as *intra-sentence relation extraction* [Bach and Badaskar, 2007]. However, relations can exist between more than two entities that appear across consecutive sentences. For example, in the text span comprising the two consecutive sentences given in LISTING 1, there exists a ternary relation response across three entities: *EGFR*, *L858E*, *gefitnib*. This relation extraction task, focusing on identifying relations between more than two entities – either appearing in a single sentence or across sentences, is known as *cross-sentence n-ary relation extraction.*

LISTING 1: TEXT SPAN OF TWO CONSECUTIVE SENTENCES

"*The deletion mutation on exon-19 of **EGFR** gene was present in 16 patients, while the **L858E** point mutation on exon-21 was noted in 10. All patients were treated with **gefitnib** and showed a partial response.*"

This paper focuses on the *cross-sentence n-ary relation extraction* task. Formally, let $\{e_1,..,e_n\}$ be the set of entities in a text span $S$ containing $t$ consecutive sentences. For example, in the text span comprising 2 sentences ($t = 2$) given in LISTING 1 above, the relation that can be extracted is that cancer patients with mutation $v$ (*EGFR*) in gene $g$ (*L858E*) demonstrated response to drug $d$ (*gefitnib*). Thus, a ternary relation response(*EGFR*, *L858E*, *gefitnib*) exists among the three entities spanning across the two sentences in LISTING 1. The entities $e_1,..,e_n$ in a text span can either appear in a single

sentence ($t = 1$) or multiple sentences ($t > 1$). Thus, given an instance defined as a combined sequence of $m$ tokens $\mathbf{x} = x_1, x_2, ..., x_m$ in $t$ consecutive sentences and a set of entities $\{e_1, .., e_n\}$, the cross-sentence $n$-ary relation extraction task is to identify the $n$-ary relation (if it exists) among the entities in $\mathbf{x}$.

Cross-sentence $n$-ary relation extraction is particularly challenging compared to intra-sentence relation extraction for several reasons. While lexico-syntactic pattern-based relation extraction methods [Hearst, 1992, Brin, 1998, Agichtein and Gravano, 2000] are useful for intra-sentence relation extraction, such pattern-based relation extraction methods cannot be readily applied to cross-sentence $n$-ary relation extraction, because it fails to match lexico-syntactic patterns across longer text spans covering multiple sentences. Although, dependency-based features [Culotta and Sorensen, 2004, Bunescu and Mooney, 2005, Fundel et al., 2006, Xu et al., 2015, Miwa and Bansal, 2016] are useful for intra-sentence relation extraction, it is not clear how best to merge dependency parse trees from different sentences to extract path-based features for cross-sentence relation extraction. Difficulties in coreference resolution and discourse analysis, further complicate $n$-ary cross-sentence relation extraction [Peng et al., 2017].

The principal challenges for cross-sentence $n$-ary relation extraction arise from: (a) difficulties in handling long-range sequences resulting from combining multiple sentences, (b) modeling the contexts of words in relation to entities across sentences, and (c) the problem of representing a variable-length text span containing an $n$-ary relation using a fixed-length representation. To address these issues, a combined model consisting of a Long Short-Term Memory unit and a Convolutional Neural Network (LSTM_CNN) that exploits both word embedding and positional embedding features, is proposed for cross-sentence $n$-ary relation extraction. The LSTM is used as the first layer to encode the combined set of sentences representing an $n$-ary relation, thereby capturing the long-range sequential information. The hidden state representation obtained from the LSTM is then used with the CNN to further identify the salient features for relation classification.

Further, although combined models of CNNs and RNNs are explored for text classification [Lai et al., 2015, Lee and Dernoncourt, 2016, Hsu et al., 2017, Zhang et al., 2016] and sentiment analysis [Wang et al., 2016], to the best of our knowledge, we are the first to propose a combined LSTM_CNN model for cross-sentence $n$-ary relation extraction.

Zhang et al. 2016 proposed a dependency sensitive convolution neural network (DSCNN) model as a general-purpose classification system, very similar to our proposed model. However, the DSCNN model does not employ position embeddings to differentiate the input words $w.r.t$ the entities, that are crucial for relation extraction. In contrast, our proposed model employs position embeddings in the combined model to achieve higher performance for $n$-ary relation extraction. Our main contributions are:

a. Propose LSTM_CNN+WF+PF model exploiting word embedding and position embedding features for cross-sentence $n$-ary relation extraction. The LSTM_CNN+WF+PF model is evaluated against baseline models such as CNN, LSTM and CNN_LSTM and show that LSTM_CNN+WF+PF significantly outperforms the baselines.

b. An evaluation of the proposed model against State-Of-The-Art (SOTA) for cross-sentence $n$-ary relation extraction on two different benchmark datasets is presented.

Results show that the proposed model significantly outperforms the current SOTA methods for cross-sentence $n$-ary relation extraction.

## 2. Related Work

Most of the studies on cross-sentence relation extraction focus on extracting binary relations present across sentences [Swampillai and Stevenson, 2010, Quirk and Poon, 2016, Moschitti et al., 2013, Nagesh, 2016]. However, recently, Peng et al. 2017 proposed graph-LSTMs to extract $n$-ary relations present in a single sentence and across sentences. Specifically, binary relations ($n$=2) and ternary relations ($n$=3) were considered. The authors noted that they could not find sufficient number of relation instances where $n > 3$ [Peng et al., 2017] suitable for experiments. The dataset developed by Peng et al. is the largest dataset currently available for $n$-ary cross-sentence relation extraction. In addition, chemical-induced disease relation extraction dataset [Li et al., 2016] provides instances with binary relations in single sentences and across two sentences. We use these two datasets in this study.

Cross-sentence relation extraction has largely used dependency features, which are useful in providing connections between sentences [Swampillai and Stevenson, 2010, Quirk and Poon, 2016, Peng et al., 2017]. Tree kernel features [Moschitti et al., 2013, Nagesh, 2016] are also used to provide more efficient and comprehensive feature sets. Recently Peng et al. 2017 proposed graph-LSTMs to capture intra-and inter-sentence dependencies for cross-sentence $n$-ary relation extraction. Although graph-LSTMs are useful in exploiting graph edges, Song et al. 2018 proposed graph-state LSTM model that uses parallel states to model each word, recurrently enriching state values via message passing. While graph-LSTMs and graph-state LSTMs are useful, creating DAGs covering words in multiple sentences is complex and error-prone. It is not obvious how to connect two parse trees; also the parse errors will compound during the graph creation step. Moreover, the co-reference resolution and discourse features do not always improve performance [Peng et al., 2017]. In contrast, we present a simple neural network-based approach that does not rely on heavy syntactic features such as dependency trees, co-reference resolution or discourse features for cross-sentence $n$-ary relation extraction.

## 3. Cross-Sentence $n$-ary Relation Extraction

The architecture of LSTM_CNN+WF+PF model for cross-sentence $n$-ary relation extraction is shown in Figure 1. The different components of the model is explained below.

### 3.1 Input Representation

The input to the LSTM_CNN+WF+PF model is the combined sequence of tokens in a text span $S$ comprising $t$ consecutive sentences where an $n$-ary relation exists between $n$ entities. The sequence of tokens is transformed into a combination of word embeddings and position embeddings as described below.

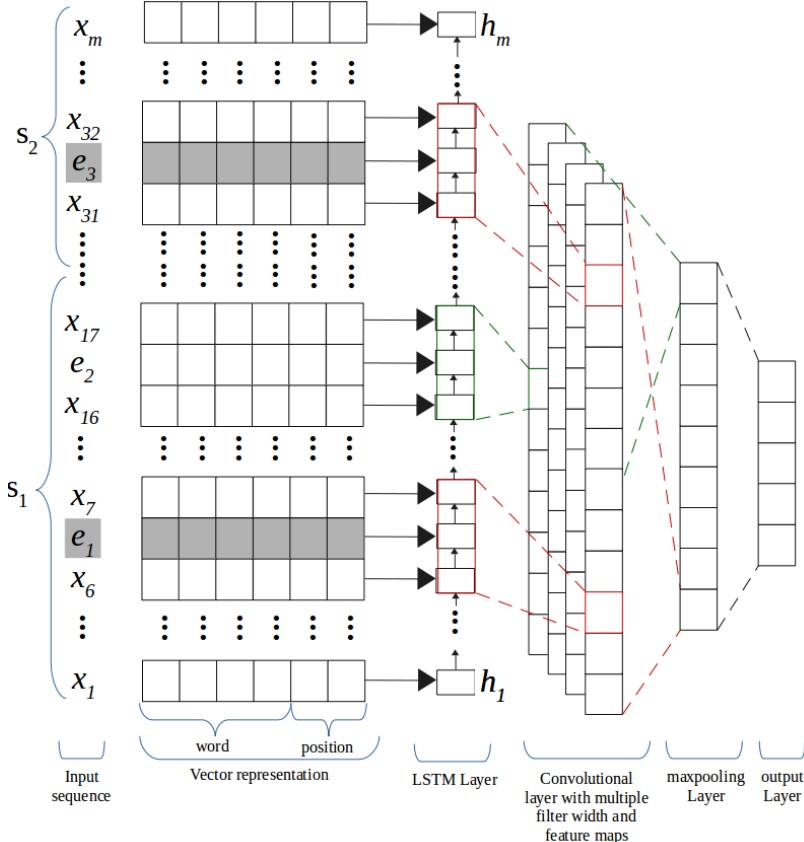

Figure 1: Architecture of the LSTM_CNN+WF+PF model for cross-sentence $n$-ary relation extraction. The input to the network is the sequence of tokens from a text span. The position features are derived for entities $e_1$ and $e_3$ (highlighted in the figure).

### 3.1.1 WORD EMBEDDINGS

The transformation of words into lower dimensional vectors are observed to be useful in capturing semantic and syntactic information about words [Mikolov et al., 2013, Pennington et al., 2014]. Thus, each of the words in the combined sequence $x = \{x_1, x_2, ..., x_n\}$ is mapped to a $k-$dimensional embedding vector using a look-up matrix $\mathbf{W} \in \mathbb{R}^{|V| \times k}$ where $|V|$ is the number of unique words in the vocabulary.

### 3.1.2 POSITION FEATURES

Following Zeng et al. 2014, Positional Features (PFs) are used to encode the position of entities for $n$-ary cross-sentence relation extraction. Given entity mentions $e_1, .., e_n$ in the sequence $x = x_1, x_2, ..., x_n$, although $n$ PFs can be defined based on $n$ entities, the proposed model, specifically considers only $e_1$ and $e_n$ to create position embeddings. The preliminary experiments demonstrated a decrease in performance with having $n$ PFs in the model. Thus, the model defines two sets of PFs $PF_1$ and $PF_n$ for the entities $e_1$ and $e_n$, respectively, as

a combination of the relative distances from the current word to the respective entity. The position embedding matrices are randomly initialised and the relative distance of words $w.r.t$ entities are transformed into real valued vectors by looking up the position embedding matrices.

Thus, the vector representation for models using position features, transforms an instance into a matrix $\mathbf{S} \in \mathbb{R}^{s \times d}$ by combining the word embeddings and position embeddings, where $s$ is the sentence length and $d = d^a + d^b \times 2$ ($d^a$ and $d^b$ are the dimensionalities of word and position embeddings, respectively).

## 3.2 LSTM Layer

Although RNNs are useful in learning from sequential data, these networks are observed to suffer from the problem of exploding or vanishing gradient, which makes it difficult for RNNs to learn long distance correlations in a sequence [Hochreiter and Schmidhuber, 1997, Hochreiter et al., 2001]. To specifically address this issue of learning long-range dependencies, LSTM [Hochreiter et al., 2001] was proposed, which maintains a separate memory cell that updates and exposes the content only when deemed necessary. Given the long-range sequential information resulting from the combined set of sentences expressing an $n$-ary relation, LSTM is an excellent choice to learn long-range dependencies. Thus, as shown in Figure 1, the transformed vector representation combining word embeddings and position features is provided as input to the LSTM layer. The LSTM units at each time step $t$ is defined as a collection of vectors in $\mathbb{R}^l$ and comprises the following components: an input gate $i_t$, a forget gate $f_t$, an output gate $o_t$, a memory cell $c_t$ and a hidden state $h_t$. $l$ is the number of LSTM units and the entries of the gating vectors $i_t$, $f_t$ and $o_t$ are in $[0, 1]$. The three adaptive gates $i_t$, $f_t$ and $o_t$ depend on the previous state $h_{t-1}$ and the current input $x_t$ (Equations 1-3). The candidate update vector $g_t$ (Equation 4) is also computed for the memory cell. The current memory cell $c_t$ is a combination of the previous cell content $c_{t-1}$ and the candidate content $g_t$, weighted respectively by the input gate $i_t$ and forget gate $f_t$ (Equation 5). The hidden state $h_t$, which is the output of the LSTM units is computed using Equation 6. $\sigma$ denotes a sigmoid function and $\odot$ denotes element-wise multiplication.

$$i_t = \sigma(\mathbf{W}_i x_t + \mathbf{U}_i h_{t-1} + b_i) \tag{1}$$

$$f_t = \sigma(\mathbf{W}_f x_t + \mathbf{U}_f h_{t-1} + b_f) \tag{2}$$

$$o_t = \sigma(\mathbf{W}_o x_t + \mathbf{U}_o h_{t-1} + b_o) \tag{3}$$

$$g_t = \tanh(\mathbf{W}_g x_t + \mathbf{U}_g h_{t-1} + b_g) \tag{4}$$

$$c_t = i_t \odot g_t + f_t \odot c_{t-1} \tag{5}$$

$$h_t = o_t \odot \tanh(c_t). \tag{6}$$

## 3.3 CNN Layer

Let $h_i \in \mathbb{R}^l$ be the $l$-dimensional hidden state vector corresponding to the $i$-th token in the combined sequence $\mathbf{x}$. The combined hidden state vectors in the sequence of length $m$ is represented as:

$$h_{1:m} = h_1 \oplus h_2 \oplus ... \oplus h_m, \tag{7}$$

where $\oplus$ denotes vector concatenation. In general, let $h_{i:i+j}$ refer to the concatenation of hidden state vectors $h_i, h_{i+1}, ..., h_{i+j}$. The convolution operation involves a filter $\mathbf{w} \in \mathbb{R}^{pl}$, which is applied to a window of $p$ hidden state vectors to generate a new feature. For instance, a feature $c_i$ is generated from a window of hidden state vectors $h_{i:i+p-1}$ calculated as follows:

$$c_i = f(\mathbf{w} \cdot h_{i:i+p-1} + b). \tag{8}$$

Here $b \in \mathbb{R}$ is the bias term and $f$ is a non-linear function such as the rectified linear unit (ReLU). This filter is applied to each possible window of hidden state vectors in the combined sequence $h_{1:p}, h_{2:p+1}, \ldots, h_{m-p+1:n}$ to produce a feature map $c \in \mathbb{R}^{m-p+1}$ given by:

$$c = [c_1, c_2, ..., c_{m-p+1}]. \tag{9}$$

Max-pooling is applied over the feature map to take the maximum value $\hat{c} = \max\{c\}$ as the feature corresponding to this particular filter. The use of multiple filters facilitates selection of the most important feature (one with the highest value) for each feature map. Finally, the use of multiple filters with varying window sizes result in a fixed length vector $\mathbf{g} \in \mathbb{R}^{fw}$, where $f$ is the number of filters and $w$ is the number of different window sizes.

## 3.4 Predicting $n$-ary Relations

The task of predicting $n$-ary relations is modeled both as a binary and multi-class classification problem. The output feature vector $\mathbf{g}$ obtained from the convolution and max-pooling operation is passed to a softmax layer, to obtain the probability distribution over relation labels. Dropout [Srivastava et al., 2014] is used on the output layer to prevent over-fitting. Thus, given a set of instances, with each instance being a text span $S_i$ comprising $t$ consecutive sentences (combined sequence of tokens $\mathbf{x} = x_1, x_2, ...x_m$), entity mentions $e_1, ..., e_n$ and having an $n$-ary relation $r$, the cross-entropy loss for this prediction is defined as follows:

$$J(\theta) = \sum_{i=1}^{s} \log p(r_i|S_i, \theta) \tag{10}$$

where $s$ indicates the total number of text spans and $\theta$ indicates the parameters of the model.

## 3.5 Implementation details

The proposed model was implemented using Tensorflow [Abadi et al., 2016] and will be made publicly available upon paper acceptance. The hyper-parameters of the models were set based on preliminary experiments using an independent development dataset. Training was performed following mini-batch gradient descent (SGD) with a batch size of 10. The models were trained for at most 30 epochs, which was sufficient to converge. The size of the hidden vectors for the LSTM was set to 300. The window sizes for the CNN was set to 3,4 and 5, and experiments were conducted with different number of filters set to 10 and 128. Word embeddings were initialised using publicly available 300-dimensional Glove word vectors trained on a 6 billion token corpus from Wikipedia and web text [Pennington et al., 2014]. The dimensions for position embeddings was set to 100 and were initialised randomly between [-0.25, 0.25].

## 4. Experiments

### 4.1 Datasets

The following datasets are used in this study.

#### 4.1.1 Quirk and Poon (qp) Dataset

The dataset[1] developed by Quirk and Poon 2016 and Peng et al. 2017 is used in this study. Distant supervision was adopted to extract relations involving *drug, gene* and *mutation* triples from the biomedical literature available in PubMed Central[2]. A *minimal span* [Quirk and Poon, 2016] was used to avoid co-occurrence of the same entity triples and to obtain spans with $\leq 3$ consecutive sentences to avoid candidates where triples are far apart in the span. A total of 59 drug-gene-mutation triples was used to obtain 3,462 ternary relation instances and 3,192 binary relation instances (involving drug-mutation entities) as positive examples. The dataset has instances with ternary and binary relations, either appearing in a single sentence or across sentences. Each instances is labeled using four labels: 'resistance', 'resistance or non-response', 'response', and 'sensitivity'. The label 'none' is used for negative instances. Negative samples were generated by randomly sampling co-occurring entity triples without known interactions, following the same restrictions used for obtaining positive samples. Negative examples were sampled to match the same number of positive samples to develop a balanced dataset.

#### 4.1.2 Chemical Induced Disease (cid) Dataset

We also use cid dataset[3] comprising binary relation instances between chemicals and related diseases. We followed the methodology of Gu et al. gu2016chemical to obtain relation instances from the corpus. A total of 1206, 1999 and 1330 positive instances were obtained for binary relations in single sentences and a total of 702, 788 and 786 positive instances were binary relations across sentences, respectively. Negative instances were created following the same restrictions, however without any known interactions between entities.

### 4.2 Evaluation Metrics

Following Peng et al. 2017, we conduct five-fold cross-validation and report average test accuracy on held-out folds using the q&p dataset. To avoid training and test contamination, held-out evaluation was conducted in each fold, based on categorizing instances related to specific entity pairs (binary relations) or entity triples (ternary relations). For example, for binary relations, the instances relating to the first 70% of the entity pairs drawn from a unique list of entity pairs was used as the training set. Instances relating to the next 10% and last 20% were used as development set and test set, respectively. For cid dataset, the Precision, Recall and F-score for test set is reported as the corpus is available as training, development and test sets. Previous studies follow a similar metric [Gu et al., 2016, 2017, Zhou et al., 2016].

---

1. http://hanover.azurewebsites.net

2. http://www.ncbi.nlm.nih.gov/pmc

3. https://github.com/JHnlp/BC5CIDTask

## 4.3 Baseline models

The proposed model is evaluated against the following baselines: a CNN model using word features (**cnn+wf**) and both word features and positional features (**cnn+wf+pf**); an LSTM model using word features (**lstm+wf**) and using both word features and positional features (**lstm+wf+pf**); a model that begins with a CNN layer followed by an LSTM layer and uses word features (**cnn_lstm+wf**) and both word features and position features (**cnn_lstm+wf+pf**); and finally a model that begins with an LSTM layer followed by a CNN layer and employs word features (**lstm_cnn+wf**).

## 4.4 Results and Discussion

The following are the results of this study.

### 4.4.1 PERFORMANCE OF THE PROPOSED MODEL.

The proposed LSTM_CNN+WF+PF model as shown in Tables 1 and 2 achieves statistically significant accuracy ($p \leq 0.05$; Friedman Test) against all baseline models, for both cross-sentence ternary and binary relation extraction on Q&P DATASET . The higher performance of the combined LSTM_CNN model against CNN and LSTM used in isolation, indicates the usefulness of combining LSTM and CNN to exploit together the strength of LSTMs to learn from long sequences (input sequences) and the ability of CNNs to identify salient features from the hidden-state output from LSTM, particularly for cross-sentence $n$-ary relation extraction. The combined model benefits from using both both word features (WF) and positional features (PF). Interestingly, the use of WF alone helps the combined model (LSTM_CNN) in achieving higher performance, particularly for binary relations (single and across sentences) and ternary relations (single sentences) (Tables 1 and 2 with $nf = 128$). However, it is the addition of PF that drastically improves the performance, by providing useful encoding of the position of words $w.r.t$ entities in the text span.

### 4.4.2 WHERE DOES LSTM_CNN MODEL SCORE?

To analyse the contribution of LSTM_CNN+WF+PF on text spans with different lengths, we divided each dataset into three groups based on the distance between entity $e_1$ and $e_n$ in the text span. Specifically, we calculated the average number of tokens ($\mu$) between $e_1$ and $e_n$ and the standard deviation ($\sigma$) over different lengths of tokens between $e_1$ and $e_n$ in the dataset. Thus, if $k$ is the total number of tokens between $e_1$ and $e_n$, the dataset was divided into the following three groups: (a) short-distance spans ($k \leq \mu - \sigma$); (b) medium-distance spans ($\mu - \sigma < k < \mu + \sigma$); (c) long-distance spans ($k \geq \mu + \sigma$). The performance of various models on the above three groups of sentences provided in Table 3, shows that LSTM_CNN+WF+PF model scores higher, particularly for medium-distance spans ($\mu - \sigma < k < \mu + \sigma$) and long-distance spans ($k \geq \mu + \sigma$). For short-distance and long-distance spans involving ternary relations across sentences, the LSTM_CNN+WF+PF model predicts ternary relations correctly for 81.3% and 82.9% spans, respectively. Similarly, the percentage of correct predictions for binary relation extraction in single sentences and across sentences is significantly higher than the performance of other models. These results clearly indicate that the combined LSTM_CNN+WF+PF model is more useful compared to CNN and

|  | single sentence | | cross sentences | |
| --- | --- | --- | --- | --- |
|  | $nf{=}10$ | $nf{=}128$ | $nf{=}10$ | $nf{=}128$ |
| CNN+WF | 72.5 | 75.5 | 75.2 | 76.3 |
| CNN+WF+PF | 73.3 | 73.9 | 78.5 | 78.7 |
| LSTM+WF† | - | 75.0 | - | 78.2 |
| LSTM+WF+PF† | - | 74.5 | - | 78.9 |
| CNN_LSTM+WF | 77.6 | 75.4 | 76.9 | 75.3 |
| CNN_LSTM+WF+PF | 72.0 | 53.0 | 76.8 | 62.6 |
| LSTM_CNN+WF | 78.3 | 78.4 | 77.5 | 78.8 |
| LSTM_CNN+WF+PF | 73.1 | **79.6*** | 80.5 | **82.9*** |

Table 1: Average test accuracy in five-fold cross-validation for *drug-gene-mutation ternary interactions* in QP DATASET. $nf$ - number of filters. † LSTM+WF and LSTM+WF+PF models does not use filters

|  | single sentence | | cross sentences | |
| --- | --- | --- | --- | --- |
|  | $nf{=}10$ | $nf{=}128$ | $nf{=}10$ | $nf{=}128$ |
| CNN+WF | 68.9 | 72.4 | 73.2 | 76.6 |
| CNN+WF+PF | 74.0 | 74.2 | 81.3 | 81.3 |
| LSTM+WF† | - | 75.4 | - | 80.3 |
| LSTM+WF+PF† | - | 74.4 | - | 80.8 |
| CNN_LSTM+WF | 71.2 | 72.3 | 76.5 | 76.5 |
| CNN_LSTM+WF+PF | 74.7 | 56.2 | 81.2 | 74.4 |
| LSTM_CNN+WF | 74.9 | 76.7 | 79.7 | 82.0 |
| LSTM_CNN+WF+PF | 85.3 | **85.8*** | 85.1 | **88.6*** |

Table 2: Average test accuracy in five-fold cross-validation for *drug-gene binary interactions* in QP DATASET. $nf$ - number of filters. † LSTM+WF and LSTM+WF+PF models does not use filters

LSTM models, particularly where the distance between $e_1$ and $e_2$ is large. In other words the combined LSTM_CNN models are more useful in extracting relations from larger spans of consecutive sentences.

Further, the highest margin between LSTM_CNN+WF+PF and the baselines is recorded for binary interactions in single sentences and across sentences with an accuracy of 85.8 and 88.6, respectively (Table 2). This is followed by ternary interactions in single sentences and across sentences with an accuracy of 79.6 and 82.9, respectively (Table 1). It is interesting to note that the average length of tokens ($\mu$) between entities in text spans in the datasets relating to binary and ternary interactions in single sentences and across sentences is of the order 19, 29, 34 and 44, respectively. Based on these results, it can be broadly concluded

| Model | $k \leq \mu - \sigma$ (%) | $\mu - \sigma < k < \mu + \sigma$ (%) | $k \geq \mu + \sigma$ (%) |
|---|---|---|---|
| *drug-gene-mutation - ternary relations - cross sentence ($\mu$=44)* | | | |
| CNN+WF | 82.9 | 74.9 | 79.8 |
| CNN+WF+PF | 84.7 | 76.5 | 80.3 |
| LSTM+WF | 46.2 | 77.0 | 79.5 |
| LSTM+WF+PF | 54.2 | 77.6 | 80.4 |
| CNN_LSTM+WF | 51.4 | 74.9 | 79.0 |
| CNN_LSTM+WF+PF | 86.2 | 74.8 | 78.8 |
| LSTM_CNN+WF | 52.0 | 76.0 | 79.1 |
| LSTM_CNN+WF+PF | 81.3 | 81.3 | 82.9 |
| *drug-gene-mutation - ternary relations - single sentence ($\mu$=34)* | | | |
| CNN+WF | 20.0 | 73.1 | 86.6 |
| CNN+WF+PF | 10.0 | 72.0 | 83.4 |
| LSTM+WF | 20.0 | 73.5 | 85.8 |
| LSTM+WF+PF | 20.0 | 73.0 | 85.6 |
| CNN_LSTM+WF | 20.0 | 76.2 | 87.3 |
| CNN_LSTM+WF+PF | 20.0 | 69.7 | 88.8 |
| LSTM_CNN+WF | 20.0 | 76.8 | 88.0 |
| LSTM_CNN+WF+PF | 20.0 | 79.5 | 86.6 |
| *drug-mutation - binary relations - cross sentence ($\mu$=29)* | | | |
| CNN+WF | 0.0 | 79.6 | 78.1 |
| CNN+WF+PF | 20.0 | 83.9 | 82.7 |
| LSTM+WF | 20.0 | 80.7 | 79.9 |
| LSTM+WF+PF | 20.0 | 81.2 | 80.5 |
| CNN_LSTM+WF | 20.0 | 78.0 | 81.3 |
| CNN_LSTM+WF+PF | 20.0 | 84.8 | 87.3 |
| LSTM_CNN+WF | 20.0 | 81.6 | 83.2 |
| LSTM_CNN+WF+PF | 20.0 | 90.9 | 90.2 |
| *drug-mutation - binary relations - single sentence ($\mu$=19)* | | | |
| CNN+WF | 16.1 | 73.5 | 66.6 |
| CNN+WF+PF | 18.4 | 74.8 | 67.3 |
| LSTM+WF | 17.6 | 77.7 | 66.5 |
| LSTM+WF+PF | 16.9 | 75.7 | 64.9 |
| CNN_LSTM+WF | 15.3 | 72.7 | 62.5 |
| CNN_LSTM+WF+PF | 19.2 | 76.8 | 65.8 |
| LSTM_CNN+WF | 16.1 | 76.4 | 67.6 |
| LSTM_CNN+WF+PF | 17.6 | 84.9 | 86.5 |

Table 3: Performance of models on different groups of sentences. $k$ - length of tokens between entities $e_1$ and $e_n$, $\mu$ average number of tokens between $e_1$ and $e_n$, $\sigma$ standard deviation over the length of tokens.

that the contribution of PF decreases with the increase in the distance between entities in the text span.

### 4.4.3 lstm_cnn vs. cnn_lstm.

The results indicate that it is more useful to start with an LSTM layer followed by CNN layer than having a CNN_LSTM model for cross-sentence $n$-ary relation extraction. As seen from Tables 1 and 2, the LSTM_CNN models perform significantly higher than CNN_LSTM models both for ternary and binary relations in single sentences and across sentences. A LSTM_CNN model is more useful in that, it initially learns from the sequential information available in the input, which is further exploited by the CNN max-pooling layer to identify salient features. However, in the CNN_LSTM model, although the use of a CNN layer with max-pooling as the fist component helps in identifying salient features from the input, the absence of sequential data from the CNN layer results in poor performance. Further, as the results show, the addition of position embeddings in the CNN_LSTM model (CNN_LSTM+WF+PF) results in poor performance in comparison to the use of word embeddings alone (CNN_LSTM+WF). This is particularly true for ternary relation extraction (Table 1). Further as seen in Table 1, the use of a higher number of filters combining word embeddings and position embeddings, dramatically lowers the performance. This indicates that position embeddings along with a higher number of filters are not useful for CNN_LSTM models. However, it is worth noting (from Table 3) that the CNN_LSTM+WF+PF model extracts ternary relations in single sentences for the higher number of long-distance spans (88.8%), indicating that CNN_LSTM models are useful in certain cases.

### 4.4.4 cnn and lstm models.

The results show that, when used in isolation, LSTM-based models are more useful for cross-sentence $n$-ary relation extraction than CNN-based models. Interestingly, using PF helps only longer sequences (accuracy of 78.9 (LSTM+WF+PF) vs. 78.2 (LSTM+WF) and 80.8 LSTM+WF+PF) vs. 80.3 (LSTM+WF+PF) scored for ternary relations in drug-mutation-gene (Table 1) and drug-mutation (Table 2), respectively). However, for shorter sequences, using PF results in a decrease in accuracy (accuracy of 74.5 (LSTM+WF+PF) vs. 75.0 (LSTM+WF) and 74.4 LSTM+WF+PF) vs. 75.4 (LSTM+WF+PF) scored for binary relations in drug-mutation-gene (Table 1) and drug-mutation (Table 2), respectively). The contribution of WF in the CNN model significantly improves with the use of higher number of filters, so much so that the model performs better than combining WF and PF. This is particularly true for extracting ternary relations in single sentences (Table 1).

### 4.4.5 $n$-positional embeddings.

Given entities $e_1, ..e_n$, the LSTM_CNN+WF+PF model employs $e_1$ and $e_n$ to create positional embeddings. However, $n$-positional embeddings can be created for each of the $n$ entities and thus, a model using $n$-positional embeddings was evaluated. The use of $n$-positional embeddiung resulted in a lower accuracy of 80.5 and 77.9 (as against 82.5 and 79.6 for $e_1$ and $e_n$) for ternary relation extraction across sentence and single sentences, respectively. This indicates that using positional embeddings for $e_1$ and $e_n$ is more useful for cross-sentence relation extraction.

4.4.6 Comparison against state-of-the-art.

**q&p dataset**. The performance of lstm_cnnw-wf+pf against different methods using q&p dataset is provided in Table 4. As seen in the Table, the lstm_cnnw-wf+pf clearly achieves a significantly higher performance against all compared SOTA methods particularly for *binary* (involving drug-mutation relations) cross-sentence relation extraction. Similarly for *ternary* (involving drug-gene-mutation relations) cross-sentence relation extraction, the lstm_cnnw-wf+pf achieves a comparable performance against SOTA methods. More specifically, the lstm_cnnw-wf+pf clearly outperform feature-based models [Quirk and Poon, 2016], bilstm [Miwa and Bansal, 2016] and tree-lstm for both *binary* and *ternary* relations, indicating the superiority of the lstm_cnnw-wf+pf model. The superior performance of the combined cnn-lstm against models used in isolation (bilstm, tree-lstm), clearly indicates that the combination of cnn and lstm is useful for better accuracy. Further, the improved performance of lstm_cnnw-wf+pf model against graph-LSTM [Peng et al., 2017] indicates that the proposed model is useful to avoid complexities arising from graph-based models such as difficulties in connecting multiple sentences and parsing errors. This is also true for graph-state LSTMs [Song et al., 2018] for *binary* cross-sentence relation extraction. For cross-sentence ternary relation extraction, the proposed model achieves a comparable performance against graph-state LSTM model [Song et al., 2018], which achieves a slightly higher performance. Given the above comparison, the strength of the proposed model comes from the fact that previous SOTA methods heavily rely on syntactic features such as dependency tress, co-reference and discourse features, which are time-consuming and less accurate particularly in the biomedical domain. However, in contrast, the proposed lstm_cnn+wf+pf model does not use any such sophisticated features, but uses simpler features such as wf and pf. The ability to provide significantly higher performance with much simpler features make the proposed lstm_cnn+wf+pf model an attractive choice for cross-sentence $n$-ary relation extraction.

**cid dataset**. The performance of the lstm_cnn+wf+pf model on cid dataset in Table 5 shows that the lstm_cnn+wf+pf model achieves a statistically significant and comparable performance for extracting binary relations from multiple sentences ($t = 2$) and single sentences ($t = 1$), respectively against supervised learning methods. The combined lstm_cnn+wf+pf model scores a higher F-score (0.63) when both single and mutliple sentences are considered.[4], providing a slight increase over using cnn and lstm separately. The cnn model Nguyen and Verspoor 2018, although achieves a high recall, suffer from lower precision. The same is true of cnn+me+pp [Gu et al., 2017] and cnn [Zhou et al., 2016]. On the other hand, lstms achieve higher precision, but perform poor on recall (lstm, lstm+svmp [Zhou et al., 2016]). The combined lstm_cnn achieves a higher precision and also does not lose on recall, resulting in a higher F-score.

## 5. Conclusion

We presented a combined lstm_cnn+wf+pf model that exploits word embeddings and position embeddings for cross-sentence $n$-ary relation extraction. The evaluation of lstm_cnn+wf+pf

---

4. The SOTA methods in Table 5 does not use any knowledge base or development set for learning the model.

| Model | Single Sent. | Cross Sents. |
|---|---|---|
| *drug-gene-mutation - ternary relations* | | |
| FEATURE-BASED | 74.7 | 77.7 |
| BILSTM | 75.3 | 80.1 |
| GRAPH LSTM-EMBED | 76.5 | 80.6 |
| GRAPH LSTM-FULL | 77.9 | 80.7 |
| BILSTM+MULTI-TASK | - | 82.4 |
| GRAPH LSTM+MULTI-TASK | - | 82.0 |
| BIDIR DAG LSTM | 75.6 | 77.3 |
| GRAPH-STATE LSTM | **80.3** | **83.2** |
| LSTM_CNN+WF+PF (proposed model) | 79.6 | 82.9 |
| *drug-mutation - binary relations* | | |
| FEATURE-BASED | 73.9 | 75.2 |
| BILSTM | 73.9 | 76.0 |
| BILSTM-SHORTEST-PATH | 70.2 | 71.7 |
| TREE-LSTM | 75.9 | 75.9 |
| GRAPH LSTM-EMBED | 74.3 | 76.5 |
| GRAPH LSTM-FULL | 75.6 | 76.7 |
| BILSTM+MULTI-TASK | - | 78.1 |
| GRAPH LSTM+MULTI-TASK | - | 78.5 |
| BIDIR DAG LSTM | 76.9 | 76.4 |
| GRAPH-STATE LSTM | 83.5 | 83.6 |
| LSTM_CNN+WF+PF (proposed model) | **85.8*** | **88.6*** |

Table 4: Average test accuracy in five-fold cross validation of the proposed model and SOTA methods on $n$-ary cross-sentence relation extraction (Q&P DATASET)

against baseline models clearly establish that combining LSTM and CNN helps in bringing together the strength of LSTMs and CNNs for cross-sentence $n$-ary relation extraction. The evaluation of LSTM_CNN+WF+PF against SOTA methods for cross-sentence $n$-ary relation extraction, clearly demonstrate the superiority of the proposed model making it an attractive solution for cross-sentence $n$-ary relation extraction.

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

| Model | P | R | F |
|---|---|---|---|
| Single sentences (text span where $t=1$) | | | |
| LINGUISTIC FEATURES [Gu et al., 2016] | 0.67 | 0.68 | 0.68 |
| CNN [Gu et al., 2017] | 0.59 | 0.55 | 0.57 |
| LSTM_CNN+WF+PF (proposed model) | **0.69** | **0.70** | **0.69** |
| Across sentences (text span where $t=2$) | | | |
| LINGUISTIC FEATURES [Gu et al., 2016] | 0.51 | 0.29 | 0.37 |
| MAXIMUM ENTROPY [Gu et al., 2017] | 0.51 | 0.07 | 0.11 |
| LSTM_CNN+WF+PF (proposed model) | **0.57** | **0.57** | **0.57***  |
| Across sentences (text span where $t\leq2$) | | | |
| LINGUISTIC FEATURES + ME [Gu et al., 2016] | 0.62 | 0.55 | 0.58 |
| CNN+ME [Gu et al., 2017] | 0.60 | 0.59 | 0.60 |
| CNN+ME+PP [Gu et al., 2017] | 0.55 | 0.68 | 0.61 |
| CNN [Zhou et al., 2016] | 0.41 | 0.55 | 0.47 |
| LSTM [Zhou et al., 2016] | 0.54 | 0.51 | 0.53 |
| LSTM+SVMP [Zhou et al., 2016] | 0.64 | 0.49 | 0.56 |
| LSTM+SVM+PP [Zhou et al., 2016] | 0.55 | 0.68 | 0.61 |
| SVM [Xu et al., 2016] | 0.55 | 0.68 | 0.61 |
| CNN | 0.54 | 0.69 | 0.61 |
| CNN+CNNCHAR | 0.57 | 0.68 | 0.62 |
| CNN+LSTMCHAR [Nguyen and Verspoor, 2018] | 0.56 | 0.68 | 0.62 |
| LSTM_CNN+WF+PF (proposed model) | **0.63** | **0.63** | **0.63** |

Table 5: Performance of LSTM_CNN+WF+PF against SOTA on CID DATASET. $t$ = number of sentences, P - precision, R - recall, F - F-score.

ACM, 2000.

Nguyen Bach and Sameer Badaskar. A survey on relation extraction. *Language Technologies Institute, Carnegie Mellon University*, 2007.

Sergey Brin. Extracting patterns and relations from the world wide web. In *International Workshop on The World Wide Web and Databases*, pages 172–183. Springer, 1998.

Razvan C Bunescu and Raymond J Mooney. A shortest path dependency kernel for relation extraction. In *Proceedings of the conference on human language technology and empirical methods in natural language processing*, pages 724–731. Association for Computational Linguistics, 2005.

Aron Culotta and Jeffrey Sorensen. Dependency tree kernels for relation extraction. In *Proceedings of the 42nd annual meeting on association for computational linguistics*, page 423. Association for Computational Linguistics, 2004.

Katrin Fundel, Robert Küffner, and Ralf Zimmer. Relexrelation extraction using dependency parse trees. *Bioinformatics*, 23(3):365–371, 2006.

Jinghang Gu, Longhua Qian, and Guodong Zhou. Chemical-induced disease relation extraction with various linguistic features. *Database*, 2016, 2016.

Jinghang Gu, Fuqing Sun, Longhua Qian, and Guodong Zhou. Chemical-induced disease relation extraction via convolutional neural network. *Database*, 2017, 2017.

Marti A Hearst. Automatic acquisition of hyponyms from large text corpora. In *Proceedings of the 14th conference on Computational linguistics-Volume 2*, pages 539–545. Association for Computational Linguistics, 1992.

Sepp Hochreiter and Jürgen Schmidhuber. Long short-term memory. *Neural computation*, 9(8):1735–1780, 1997.

Sepp Hochreiter, Yoshua Bengio, Paolo Frasconi, Jürgen Schmidhuber, et al. Gradient flow in recurrent nets: the difficulty of learning long-term dependencies, 2001.

Shiou Tian Hsu, Changsung Moon, Paul Jones, and Nagiza Samatova. A hybrid cnn-rnn alignment model for phrase-aware sentence classification. In *Proceedings of the 15th Conference of the European Chapter of the Association for Computational Linguistics: Volume 2, Short Papers*, volume 2, pages 443–449, 2017.

Siwei Lai, Liheng Xu, Kang Liu, and Jun Zhao. Recurrent convolutional neural networks for text classification. In *AAAI*, volume 333, pages 2267–2273, 2015.

Ji Young Lee and Franck Dernoncourt. Sequential short-text classification with recurrent and convolutional neural networks. *arXiv preprint arXiv:1603.03827*, 2016.

Jiao Li, Yueping Sun, Robin J Johnson, Daniela Sciaky, Chih-Hsuan Wei, Robert Leaman, Allan Peter Davis, Carolyn J Mattingly, Thomas C Wiegers, and Zhiyong Lu. Biocreative v cdr task corpus: a resource for chemical disease relation extraction. *Database*, 2016, 2016.

Tomas Mikolov, Kai Chen, Greg Corrado, and Jeffrey Dean. Efficient estimation of word representations in vector space. *arXiv preprint arXiv:1301.3781*, 2013.

Makoto Miwa and Mohit Bansal. End-to-end relation extraction using lstms on sequences and tree structures. *arXiv preprint arXiv:1601.00770*, 2016.

Alessandro Moschitti, Siddharth Patwardhan, and Chris Welty. Long-distance time-event relation extraction. In *IJCNLP*, pages 1330–1338, 2013.

PC Nagesh. Exploiting tree kernels for high performance chemical induced disease relation extraction. In *4TH ANNUAL DOCTORAL COLLOQUIUM*, page 15, 2016.

Dat Quoc Nguyen and Karin Verspoor. Convolutional neural networks for chemical-disease relation extraction are improved with character-based word embeddings. *arXiv preprint arXiv:1805.10586*, 2018.

Nanyun Peng, Hoifung Poon, Chris Quirk, Kristina Toutanova, and Wen-tau Yih. Cross-sentence n-ary relation extraction with graph lstms. *Transactions of the Association for Computational Linguistics*, 5:101–115, 2017.

Jeffrey Pennington, Richard Socher, and Christopher Manning. Glove: Global vectors for word representation. In *Proceedings of the 2014 conference on empirical methods in natural language processing (EMNLP)*, pages 1532–1543, 2014.

Chris Quirk and Hoifung Poon. Distant supervision for relation extraction beyond the sentence boundary. *arXiv preprint arXiv:1609.04873*, 2016.

Linfeng Song, Yue Zhang, Zhiguo Wang, and Daniel Gildea. N-ary relation extraction using graph state lstm. *arXiv preprint arXiv:1808.09101*, 2018.

Nitish Srivastava, Geoffrey E Hinton, Alex Krizhevsky, Ilya Sutskever, and Ruslan Salakhutdinov. Dropout: a simple way to prevent neural networks from overfitting. *Journal of machine learning research*, 15(1):1929–1958, 2014.

Kumutha Swampillai and Mark Stevenson. Inter-sentential relations in information extraction corpora. In *LREC*, 2010.

Xingyou Wang, Weijie Jiang, and Zhiyong Luo. Combination of convolutional and recurrent neural network for sentiment analysis of short texts. In *Proceedings of COLING 2016, the 26th International Conference on Computational Linguistics: Technical Papers*, pages 2428–2437, 2016.

Jun Xu, Yonghui Wu, Yaoyun Zhang, Jingqi Wang, Hee-Jin Lee, and Hua Xu. Cd-rest: a system for extracting chemical-induced disease relation in literature. *Database*, 2016, 2016.

Yan Xu, Lili Mou, Ge Li, Yunchuan Chen, Hao Peng, and Zhi Jin. Classifying relations via long short term memory networks along shortest dependency paths. In *EMNLP*, pages 1785–1794, 2015.

Daojian Zeng, Kang Liu, Siwei Lai, Guangyou Zhou, Jun Zhao, et al. Relation classification via convolutional deep neural network. In *COLING*, pages 2335–2344, 2014.

Rui Zhang, Honglak Lee, and Dragomir Radev. Dependency sensitive convolutional neural networks for modeling sentences and documents. *arXiv preprint arXiv:1611.02361*, 2016.

Huiwei Zhou, Huijie Deng, Long Chen, Yunlong Yang, Chen Jia, and Degen Huang. Exploiting syntactic and semantics information for chemical–disease relation extraction. *Database*, 2016, 2016.