# OpenReview forum: "Combining Long Short Term Memory and Convolutional Neural Network for Cross-Sentence n-ary Relation Extraction"
_AKBC.ws/2019/Conference — AKBC 2019_

### Official Review · AnonReviewer1 · 2018-12-20
**Good paper with small modeling improvements but thorough evaluations**

**Rating:** 7
**Confidence:** 4

**Review:**

The paper presents a method for n-ary cross sentence relation extraction.
Given a list of entities, and a list of sentences the task is to identify which relation (from a predefined list) is described between the entities in the given sentences.
The proposed model stacks CNN on LSTM to get long range dependencies in the text, and shows to be effective, either beating or equalling the state-of-the-art on two datasets for the task.

Overall, I enjoyed reading the paper, and would like to see it appear in the conference.
While the proposed model is not very novel, and was shown effective on other tasks such as text classification or sentiment analysis, this is the first time it was applied for this specific task.
In addition, I appreciate the additional evaluations, which ablate the different parts of the model, analyze its performance by length between entities, compare it with many variations as baselines and against state-of-the-art for the task.

My main comments are mostly in terms of presentation - see below.


Detailed comments:

In general, I think that wording can be tighter, and some repetitive information can be omitted. For example, Section 4 could be condensed to highlight the main findings, instead of splitting them across subsections.
I think that Section 3.1.2 (“Position Features”) would benefit from an example showing an input encoding.
Table 5 shows up in the references.

Minor comments and typos:

Text on P. 9 overflows the page margins.
I think that Table 3 would be a little easier to read if the best performance in each column were highlighted in some manner.
Section 2, p. 3: “mpdel” -> model.
Perhaps using “Figure 1” instead of “Listing 1” is more consistent with *ACL-like papers?

---

> ### Author Response · Authors · 2019-02-01
> **Answers to comments**
>
> We thank the reviewer for the comments. The following is our response.
>
> Q1: In general, I think that wording can be tighter, and some repetitive information can be omitted. For example, Section 4 could be condensed to highlight the main findings, instead of splitting them across subsections.
> Ans: Since we wanted to discuss various aspects of the model, we have used different sub-sections. The title of the sub-section indicates the key aspect of the model, that we wanted to highlight and discuss. We feel that condensing it into one section, can result in difficulty in reading the paper.
>
> Q2: I think that Section 3.1.2 (“Position Features”) would benefit from an example showing an input encoding.
> Ans: We have updated Section 3.1.2 to provide an example of position embedding.
>
> Q3: Table 5 shows up in the references.
> Ans: Corrected. Table 5 is moved to the next page
>
> Q4: Typos and margins
> Ans: The typos and margins are corrected.

---

### Official Review · AnonReviewer2 · 2019-01-04
**Nice experimental study**

**Rating:** 7
**Confidence:** 4

**Review:**

The paper addresses cross-sentence n-ary relation extraction. The authors propose a model consisting of an LSTM layer followed by an CNN layer and show that it outperforms other model choices. The experiments are sound and complete and the presented results look convincing. The paper is well written and easy to follow.
In total, it presents a nice experimental study.

Some unclear issues / questions for the authors:
- "The use of multiple filters facilitates selection of the most important feature for each feature map": What do you mean with this sentence? Don't you get another feature map for each filter? Isn't the use of multiple filters rather to capture different semantics within the sentence?
- "The task of predicting n-ary relations is modeled both as a binary and multi-class classification problem": How do you do that? Are there different softmax layers? And if yes, how do you decide which one to use?
- Table 1/2: How can you draw conclusions about the performance on binary and ternary relations from these tables? I can only see the distinction of single sentence and cross sentence there.
- Table 3: The numbers for short distance spans (mostly 20.0) look suspicious to me. What is the frequency of short/medium/long distance spans in the datasets? Are they big enough to be able to draw any conclusions from them?
- You say that CNN_LSTM does not work because after applying the CNN all sequential information is lost. But how can you apply an LSTM afterwards then? Is there any recurrence at all? (The sequential information would not be lost after the CNN if you didn't apply pooling. Have you tried that?)
- Your observation that more than two positional embeddings decrease the performance is interesting (and unexpected). Do you have any insights on this? Does the model pay attention at all to the second of three entities? What would happen if you simply deleted this entity or even some context around this entity (i.e., perform an adversarial attack on your model)?

Other things that should be improved:
- sometimes words are in the margin
- there are some typos, e.g., "mpdel", "the dimensions ... was set... and were initialised", "it interesting"

---

> ### Author Response · Authors · 2019-02-01
> **Clarification for some unclear issues**
>
> We thank the reviewer for the comments.
>
> Q1. - "The use of multiple filters facilitates selection of the most important feature for each feature map": What do you mean with this sentence? Don't you get another feature map for each filter? Isn't the use of multiple filters rather to capture different semantics within the sentence?
> Ans: It is true that the use of multiple features facilitates capturing different semantics within the sentence. We have corrected this section on max-pooling in the paper.
>
> Q2.  "The task of predicting n-ary relations is modeled both as a binary and multi-class classification problem": How do you do that? Are there different softmax layers? And if yes, how do you decide which one to use?
> Ans: The task of predicting n-ary relations as a binary and multi-classification problem is specific to the datasets that are used in the paper. While using the Peng et al. (2016) dataset, we have a multi-class classification problem. However, when we are working with Chemcial-induced dataset, we have a binary classification problem, to predict whether their exists a binary relation between the entities both in single sentence and across sentences. Therefore, when working with Peng et al dataset, we use softmax function with categorical cross entropy loss to output probability over the five output classes (resistance; resistance or no-response; response; sensitivity; and none) and employ softmax layer with binary cross-entropy loss to predict probability over two classes.
>
> Q3:  Table 1/2: How can you draw conclusions about the performance on binary and ternary relations from these tables? I can only see the distinction of single sentence and cross sentence there.
> Ans: As indicated in the caption for Tables 1 and 2, while Table 1 specifically deals with ternary relations involving ternary relations between drug-gene-mutation, Table 2 deals with binary relations involving drug-gene entities. However, these binary and ternary relations exists both in single sentences and across sentences. Thus the distinction between single and across sentences is made. Given this aspect, the conclusions are drawn for the performance of binary and ternary relations in single sentences and across sentences.
>
> Q4: - Table 3: The numbers for short distance spans (mostly 20.0) look suspicious to me. What is the frequency of short/medium/long distance spans in the datasets? Are they big enough to be able to draw any conclusions from them?
> Ans: The reviewer is right in noting that there are very few instances for short distance spans. This is also the reason why the performance of different models are lower for short distance spans, compared to medium and long distance spans.
>
> Q5: - You say that CNN_LSTM does not work because after applying the CNN all sequential information is lost. But how can you apply an LSTM afterwards then? Is there any recurrence at all? (The sequential information would not be lost after the CNN if you didn't apply pooling. Have you tried that?)
> Ans: We had conducted experiments removing the max-pooling layer and passing the features to an LSTM layer. However, this did not help in improving the performance.
>
> Q6: Your observation that more than two positional embeddings decrease the performance is interesting (and unexpected). Do you have any insights on this? Does the model pay attention at all to the second of three entities? What would happen if you simply deleted this entity or even some context around this entity (i.e., perform an adversarial attack on your model)?
> Ans: Given the experimental results, we observe that adding positional embeddings for more than two entities results in a decrease in the performance. We have not conducted further experiments such as removing the second entity or context around the second entity. Removing the second entity or the context around it would certainly result in a poor performance as we would be disturbing the sequential information. However, it is worthwhile to conduct such experiments and intend to do it in the future.
>
> Q7: Typos
> Ans: Thanks for identifying the typos. The typos are corrected.

---

### Official Review · AnonReviewer3 · 2019-01-10
**Review of Combining Long Short Term Memory and Convolutional Neural Network for Cross-Sentence n-ary Relation Extraction**

**Rating:** 6
**Confidence:** 4

**Review:**

The paper presents an approach to cross-sentence relation extraction that combines LSTMs and convolutional neural network layers with word and position features.  Overall the choices made seem reasonable, and the paper includes some interesting analysis / variations (e.g., showing that an LSTM layer followed by a CNN is a better choice than the other way around).

Evaluation is performed on two datasets, Quirk and Poon (2016) and a chemical induced disease dataset.  The paper compares a number of model variations, but there don't appear to be any comparisons to state-of-the-art results on these datasets.  The paper could benefit from comparisons to SOTA on these or other datasets.

---

> ### Author Response · Authors · 2019-01-24
> **Comparison with state-of-the-art results**
>
> We thank the reviewer for the comments.
>
> The paper, in addition to evaluation of number of model variations, provides a comprehensive evaluation of comparing the proposed model against state-of-the-art results on both datasets. The comparison of results for Quirk and Poon Dataset is provided in Table 4 with the explanation provided in Section 4.4.6. Similarly, the comparison of results for Chemcial-induced dataset is provided in Table 5 and the explanation is provided in Section 4.4.6. As explained in Section 4.4.6, the proposed model clearly outperforms the state-of-the-art results on both datasets.
>
> Further, since this research specifically considers n-ary relation extraction, these two datasets were considered which involves binary and ternary relation instances. This is the reason we do not perform evaluation on standard intra-sentence relation extraction datasets such as Semeval 2010 Task 8 dataset. Further, this is inline with previous work (Peng et al., 2016) who do not consider evaluation on Semeval 2010 Task 8 relation extraction dataset.

---

### Meta-Review · Area_Chair1 · 2019-02-11
**Simple method that works well on interesting problem**

**Recommendation:** Accept (Poster)
**Confidence:** 4

**Metareview:**

The presented cross sentence relation extraction method is simple but well motivated. The experiments show that it works well, setting a new SOTA on the two relevant benchmarks. The  ablations and extended analyses are also well done and extensive. Overall, this is a clear paper with a solid contribution that we would all learn something by reading.

---

### Decision · Program_Chairs · 2019-02-15
**AKBC 2019 Conference Decision**

Accept